# In-hospital areas with distinct maintenance and staff/patient traffic have specific microbiome profiles, functions, and resistomes

Stefanie Duller,[1] Christina Kumpitsch,[1,2] Christine Moissl-Eichinger,[1,2] Lisa Wink,[1] Kaisa Koskinen Mora,[1] Alexander Mahnert[1,2]

**ABSTRACT** Hospitals are subject to strict microbial control. Stringent cleaning and confinement measures in hospitals lead to a decrease in microbial diversity, but an increase in resistance genes. Given the rise of antimicrobial resistances and healthcare-associated infections, understanding the hospital microbiome and its resistome is crucial. This study compared the microbiome and resistome at different levels of confinement (CL) within a single hospital. Using amplicon sequencing, shotgun metagenomics, and genome/plasmid reconstruction, we demonstrate that microbial composition differs in a stable way between the CLs and that the most restrictive confinement level CL1 had the lowest microbial but the highest functional diversity. This CL also exhibited a greater abundance of functions related to virulence, disease, defense, and stress response. Comparison of antibiotic resistance also showed differences among CLs in terms of the selection process and specific mechanisms for antimicrobial/antibiotic resistance. The resistances found in the samples of CL1 were mostly mediated via antibiotic efflux pumps and were mainly located on chromosomes, whereas in the other, less restrictive CL antibiotic resistances were more present on plasmids. This could be of particular importance for patient-related areas (CL2), as the potential spread of antibiotic resistances could be a major concern in this area. Our results show that there are differences in the microbiome and resistome even within a single hospital, reflecting room utilization and confinement. Since restrictive confinement selects for resistant microorganisms, strategies are required to deepen our understanding of dynamic processes of microbiome and resistome within hospital environments.

**IMPORTANCE** Effective measures to combat antibiotic resistances and healthcare-associated infections are urgently needed, including optimization of microbial control. However, previous studies have indicated that stringent control can lead to an increase in the resistance capacities of microbiomes on surfaces. This study adds to previous knowledge by focusing on the conditions in a single hospital, resolving the microbiomes and their resistomes in three different confinement levels (CL): operating room, patient-related areas, and non-patient-related areas. We were able to identify stable key taxa; profiled the capacities of taxa, functions, and antimicrobial resistances (AMR); and reconstruct genomes and plasmids in each CL. Our results show that the most restrictive CL indeed had the highest functional diversity, but that resistances were mostly encoded on chromosomes, indicating a lower possibility of resistance spread. However, clever strategies are still required to strike a balance between microbial control and selective pressures in environments where patients need protection.

**KEYWORDS** microbiome, resistome, hospital microbiome

Address correspondence to Kaisa Koskinen Mora, kaisa454@gmail.com, or Alexander Mahnert, alexander.mahnert@medunigraz.at.

The authors declare no conflict of interest.

See the funding table on p. 18.

Hospital facilities and patient rooms are subject to more stringent microbial monitoring and management as crucial procedures for preventing infections (1–4). These cleaning and disinfection procedures, as well as the spatial confinement itself, result in a higher proportion of human-associated microorganisms, but a lower overall microbial diversity (5–7). Comparing the microbiome of uncontrolled buildings (e.g., private homes) with that of controlled buildings (e.g., intensive care unit, ICU) in our previous work revealed that controlled environments have higher diversities of functions related to resistance, virulence, disease, and defense. In addition, bacteria from this controlled environment encoded a higher diversity of genes involved in multi-drug efflux, leading to antibiotic/antimicrobial resistances (AMR) (5). AMR is a serious threat to human health and one of the most crucial problems we face nowadays (8–10). AMR leads to a less effective treatment of common infections that can become life-threatening and also affect the ability to perform surgical procedures and other medical treatments (9, 10). Another substantial global problem is the so-called healthcare-associated infections (HAIs) (8, 11, 12), which are infections that patients develop in a hospital or other healthcare facility. They occur within 48 hours of admission or 30 days after patient discharge (11, 12). Despite stringent measures to reduce the risk of infection, HAIs continue to increase. In addition to the rise of HAIs, it has been shown that already one of three HAI-associated bacteria bears one or several antibiotic resistances (13). HAI caused by antibiotic-resistant bacteria are more difficult to treat, and these infections are associated not only with increased hospitalization but also with increased morbidity and mortality rates (9, 10, 14, 15). In Europe, HAIs kill more than 91,000 patients each year (16), and according to the European Centre for Disease Prevention and Control, HAIs cause more deaths than any other infectious disease under their observation (13), and the Agency for Healthcare Research and Quality rated HAIs among the top 10 causes of death in the United States (17).

According to a comprehensive global assessment of AMR burden, 4.95 million deaths were associated with AMR bacteria (10). These numbers highlight the importance of advancing research on this topic to better understand the dynamics of the hospital microbial community and any associated (antibiotic) resistance genes, the so-called resistome, to protect patients and facilitate safe recovery. In this study, we deepened our analysis of the hospital microbiome and hypothesized that different levels of confinement result in clearly distinguishable microbiome compositions and functions, even within a single hospital. To this end, we examined different surfaces in a hospital and assigned them to different confinement levels (CLs).

These CLs were assigned according to room utilization (patient-related and non-patient-related) and accessibility (restricted or accessible). CL1, an operating room (highly restricted) was allocated as the most confined area, followed by CL2, patient-related areas in ICU department (restricted), and CL3, non-patient-related areas in ICU (not or only partially restricted). In these areas, samples were collected from different rooms and sampling sites, and 16S rRNA gene sequencing and shotgun metagenomics were performed, including not only functional analysis but also reconstruction of genomes and plasmids. Since viable microbes are of greater concern regarding infection risk and patient protection, we additionally estimated growth rates and differentiated our analysis for intact microbes. The results may help us deepen our understanding of the interplay of confinement, microbiome composition, and resistome in the hospital environment. Furthermore, the obtained insights may allow us to develop new strategies to ensure safe patient recovery and reduce infection risk and microbial resistances.

## MATERIALS AND METHODS

### Study set up

The study was conducted in two different departments at the State Hospital in Graz (Austria), namely the intensive care unit (ICU) of internal medicine and an operating room for thoracic surgery. The ICU and the operating room sampling were assigned to

three confinement levels (CL) with respect to their accessibility (restricted or accessible) and room utilization (patient-related and non-patient-related areas).

Accordingly, the operating room (CL1) was allocated as the most confined area (highly restricted), followed by patient- (CL2, restricted) and non-patient-related areas (CL3, not or only partially restricted) in the ICU. The patient-related areas included an isolation room, airlock, examination room, and a two-bed patient room, whereas the non-patient-related areas contained nurse station, waiting area, and toilet. A variety of surfaces were sampled in these departments, including floors, sinks, door handles, work desks, bed frames, keyboards, medical carts, and toilet flush buttons, similar to the sampling sites of previous studies of hospital microbiomes (8, 18–23). A more detailed picture of the various locations and all appertaining sampling sites of the ICU and operating room are listed in Table S1 and shown in Fig. S1. These departments were sampled at three time points in 2018; the exact sampling dates are shown in Table S2.

## Sampling

The floor samples were collected using DNA-free pre-moistened wipes (Sterile WipeTM LP, Texwipe, Kernersville, USA). The wipes were baked at 170°C for 24 hours to destroy any remnants of DNA, moistened with 15 mL of sterile DNA-free water (LiChrosolv grade water, Merck KGaA, Darmstadt, GER), and autoclaved (121°C, 20 min) in sterile 50 mL tubes (reaction tubes 50 mL, Sarstedt AG & Co, Nuembrecht, GER). For amplicon analysis, approximately 1 $m^2$ of the floor was sampled in the different departments. To obtain sufficient DNA for metagenomic analysis, the rest of the accessible floor was wiped. In general, wipe samples were obtained by placing the wipe flat on the area to be sampled and rubbing the surface with uniform pressure in three directions (horizontal, vertical, and diagonal).

All samples were collected using sterile DNA-free gloves (G3 Sterile Sterling Nitrile Gloves, Kimberly Clark Dallas, USA), and a new pair of gloves was used for each individual sample.

All other surfaces of the ICU department and operating room were sampled using a sterile swab (BD BBL™ Culture Swabs™ EZ, Copan, ITA). For these sampling sites, the swabs were pre-moistened with 0.9% DNA-free NaCl (wt/vol) and rubbed over the surfaces (approximately 10 $cm^2$) with rotating movements and uniform pressure in horizontal, vertical, and diagonal directions. Field control samples were taken during all sampling events to estimate the effect of possible contamination on equipment and samples. To control for the sterility of the samples collected with wipes, a field control was taken by opening the 50 mL reaction tube, opening the wipe, and returning it into the reaction tube; for swab samples, the DNA-free NaCl used to moisten the swabs was used as a field control. All samples were placed on ice immediately after collection until further processing.

## Sample processing

Molecular samples were collected both for direct DNA extraction (no pre-treatments) and propidium monoazide (PMA) treatment prior to DNA extraction. Either two side-by-side swab samples (DNA and PMA) or one wipe sample (split into samples without pre-treatment and PMA pre-treatment after wipe extraction) was collected from each sampling site.

For wipe extraction, samples were transferred to DNA-free wide-neck flasks that had been baked at 250°C for 24 hours and filled with 100 mL sterile 1× PBS buffer.

Samples were shaken for 1 minute and sonicated for 2 minutes with 240 W and a frequency of 40 kHz. Subsequently, the entire volume of the samples was concentrated by centrifugation (4,000 rpm, 5 minutes, +4°C) using UV-sterilized Amicon filters (Amicon Ultra 15 mL 50 K NMGG, Merck Millipore Ltd, IRL). After enrichment of cells and macromolecules larger than 3 kDa, samples for amplicon sequencing were divided into two reaction tubes (Eppendorf tubes 1.5 mL, Eppendorf AG, GER): one for direct DNA extraction, and one for PMA treatment prior to DNA extraction. DNA (swab and wipe)

samples and the samples for shotgun metagenomics were frozen directly at −80°C until DNA extraction. PMA treatment was additionally performed to mask DNA from dead microorganisms/free DNA (24). For this purpose, 50 µM PMA (PMA™ dye, Biotium Inc., USA) was added directly to the samples (for swab samples, 250 µL of DNA-free NaCl was added beforehand); then, the samples were vortexed and incubated for 10 minutes in the dark at room temperature.

Afterward, the samples were light-exposed using a PMA-Lite LED photolysis device (465–475 nm) (PMA-Lite™, Biotum Inc., USA) for 15 minutes according to the manufacturer's instructions. After treatment, the PMA samples were also frozen at −80°C until further processing.

## DNA extraction, shotgun metagenomics, and 16S rRNA gene amplification

DNA extraction was performed using the DNeasy PowerSoil Kit (QIAGEN GmbH, GER) according to the manufacturer's instructions with following deviations: instead of 250 mg samples, either the swab (DNA sample), the swab with 250 µL DNA-free NaCl (PMA samples), or 250 µL of the wipe samples (shotgun metagenomics, DNA, and PMA samples) was added to the PowerBead Tubes, and the bead beating step was performed twice for 30 seconds at 6,400 rpm using the MagNA Lyser device (Roche Diagnostics GmbH, GER). Between the two bead-beating steps, samples were cooled on ice.

For each DNA extraction run, a kit control was processed with the samples as an extraction blank. DNA was quantified using Nanodrop (Nanodrop 2000c spectrophotometer, Bio-rad Laboratories Inc., USA). After DNA extraction, the samples were stored at −20°C until further use. Samples for shotgun sequencing were sent to Macrogen (Seoul, South Korea) for library preparation (TruSeq DNA Nano Kit) and sequencing (HiSeq X with 2 × 150 bp; 600 Mio reads per lane). The variable region V4 of the 16S rRNA gene was amplified by polymerase chain reactions (PCR). For this purpose, 10–20 ng of template DNA, the universal primers 515F (5`- GTGYCAGCMGCCGCGGTA- 3`) and 806R (5`- GGGACTACNVGGGTWTCTATT-3`) (25), and Ex Taq DNA polymerase (Takara Bio Inc., JPN) were used for each reaction mixture. Cycling conditions consisted of an initial denaturation at 94°C for 3 min, followed by 35 cycles of denaturation at 94°C for 45 s, annealing at 50°C for 60 s, extension at 72°C for 90 s, and a final extension at 72°C for 10 min (26).

Amplicon samples were taken to the Core Facility Molecular Biology at the Center of Medical Research (Graz, AUT) for library construction and next-generation sequencing (Illumina MiSeq) (27). Briefly, prior sequencing, the PCR products were normalized with a SequalPrep™ Normalization Plate (Life Technologies™, Carlsbad, US), and the samples were indexed with unique barcode sequences using index PCR. Samples were then purified using the QIAquick Gel Extraction Kit (Quiagen) and validated and quantified using the Promega Quantus™ device and Agilent 2100 Bioanalyzer (Agilent, Santa Clara, US) (27).

## Data analysis

### Amplicons

The obtained raw data were processed using QIIME2 (Versions 2020.2 - 2021.2) according to the tutorials provided by the QIIME2 developers (https://docs.qiime2.org/) (28). After importing the demultiplexed paired-end fastq data, sequences were filtered and denoised using DADA2 and trimmed to a length of 300–350 bp to obtain good quality scores (bigger than 30) (29). A taxonomic assignment was then performed using a classifier trained on the 16S rRNA gene reference sequences from the SILVA database (version 138).

As mentioned previously, field control samples were collected at every sampling event, and control samples were processed in parallel at each sample processing step to control for potential contaminants in the reagents used and on the sampling equipment.

To identify potential contaminants from the processed controls, the R package decontam was used as described before (https://github.com/benjjneb/decontam) (30).

The threshold for classifying features as potential contaminants was set to 0.5. In this way, 492 of 27,493 features were removed from the data set. Sequences assigned to mitochondrial or chloroplast signatures were also removed by taxonomy-based filtering using QIIME2 (version 2021.2) as described by the developers (https://docs.qiime2.org/2022.11/tutorials/filtering/). All subsequent analyses were performed on the cleaned and normalized data set.

## Shotgun metagenomics

Raw sequence data were analyzed using FastQC version 0.11.8 (31) and trimmed with Trimmomatic version 0.38 (32) according to a minimum Phred score of 20 within a sliding window of 5 bases, a minimum sequence length of 50 bp, as well as the removal of overrepresented sequences identified by FastQC.

We used two approaches to analyze shotgun data sets from microbial communities and their functions on hospital surfaces in detail: (i) a read-based metagenome analysis that provides an assembly-free perspective on the taxonomy and function of the hospital indoor microbiome and (ii) a genome-based metagenome analysis that provides information on assembled contigs and draft-binned genomes and their functions, including a focus on antibiotic resistances and virulence factors, comparative genomics based on their protein inventory, and inferring their activity from predicted growth rates.

## Read-based shotgun metagenomics analysis

The quality-trimmed reads were annotated using Diamond 0.9.25 (33) according to blastX searches against NCBInr database (from October 2019). Resulting m8 files were imported and visualized with MEGAN 6.12.3 (34), MEGAN Community Edition - Interactive Exploration and Analysis of Large-Scale Microbiome Sequencing Data) using default settings (LCA algorithm: naive; read assignment mode: readCount; Min Score: 50, Max Expected: 0.01, Top Percent: 10, Min Support Percent: 0.05) to assess the abundance of specific taxa (prot_acc2tax-Jul2019.abin) and functions (acc2seed-May2015XX.abin) in studied metagenomes (https://software-ab.cs.uni-tuebingen.de/download/megan6/old.html). To assess which genes and functions were carried by potentially mobile elements, we applied PlasFlow v1.1 (35) to predict plasmid sequences within the metagenomic data.

## Genome-based metagenome analysis

The quality-trimmed reads were assembled with Megahit v1.1.3 in meta-sensitive mode (36), and assembled contigs were binned using MaxBin v2.2.4 (37).

Quality in terms of completeness, contamination, and strain heterogeneity of the binned genomes was assessed using CheckM (38), and taxonomic affiliations were determined with GTDBtk v1.2.0 and database release 89 (39). Only MAGs with >90% completeness and <25% contamination were selected for further analysis. Good quality MAGs were annotated using the Magnifying Genomes (MaGe) annotation platform in MicroScope (40).

To estimate whether bacteria were active on hospital surfaces at the time of sampling, growth rates of MAGs were estimated using GRiD v1.3 (multiplex mode, coverage cutoff >= 0.2) (41). The relative abundance of each quality MAG was assessed with CoverM v0.6.1, and the minimally covered fraction was set to 80% (https://github.com/wwood/CoverM).

A protein catalog of all quality MAGs based on EggNOG mapper results (exported from MaGe) was created and weighted with the mean relative abundance of each MAG per confinement level (CL).

Antibiotic resistance genes and virulence factors on contigs, MAGs, or plasmids were profiled using abricate v1.0.1 (https://github.com/tseemann/abricate) using the following databases: CARD (42), VFDB (43), Resfinder (44), PlasmidFinder (45), ARG-ANNOT (46), EcOH (47), MEGARes 2.0 (48), as well as NCBI AMRFinderPlus (49).

### Normalization, statistics, and visualization

Data were normalized using SRS (scaling with ranked subsampling) (50). For amplicon data, a scaling factor of 1,000 and for the shotgun metagenomic gene-centric data, a scaling factor of 178,000 for taxa and 328,000 for SEED functions was chosen. All subsequent analyses were performed with the normalized data.

Both amplicon and shotgun metagenomic data were visualized using the R libraries (ggplot2, tidyverse, scales, reshape2, lattice, microbiome, knitr, ggpubr, and phyloseq) for alpha and beta diversity plots, boxplots, and heat maps.

For creating the cladograms RAWGraphs (https://github.com/rawgraphs) was used and also for boxplots for GRiD value comparison in supplemental material. In addition to RAWGraphs, supplementary figures for alpha, beta diversity, and relative abundance of metagenomic data were generated with the R package Microbiome Explorer (51).

The summary flowchart of the obtained data was generated using (https://app.dia-grams.net/). Differential abundance between confinement levels (CLs) was tested with the QIIME2 plugins aldex2 (52), ANCOM (53), and ANCOM-BC (54) and the R packages ANCOM2 (https://github.com/FrederickHuangLin/ANCOM-Code-Archive) and MaAsLin2 (https://github.com/biobakery/Maaslin2 (55), including time as a fixed effect and different room locations as a random effect where applicable.

## RESULTS

### Study overview

To investigate associations between confinement levels (CLs) and microbiome composition and function within a single hospital, 220 samples were collected from an intensive care unit (ICU) and a surgical operating room. Sampling areas were divided into different CLs with respect to room utilization and accessibility (staff and patient traffic).

Accordingly, the operating room was classified as the most restricted area (CL1), followed by patient-related areas (restricted, CL2) and non-patient-related areas (unrestricted or partially restricted, CL3) in the ICU. In total, eight rooms were sampled, and 22 different surfaces, such as sinks, bed frames, medical carts, etc., were examined (full details are given in Materials and Methods chapter and are listed and visualized in Table S1 and Fig. S1). Samples were collected at three time points for 16S rRNA gene amplicon analyses and shotgun metagenomics; the sampling dates are listed in Table S2.

An overview of the obtained data and the principle workflows of data processing can be found in Fig. S2. Only floor samples were considered for shotgun metagenomics due to the required biomass for proper shotgun library preparation. A subset of samples was treated with PMA (propidium monoazide, masking freely accessible DNA from dead microorganisms) for comparison with growth rate index measures (GRiD).

### Confinement levels are longitudinally and stably reflected in the microbiome profiles

CL1 (operating room), the most confined area, was characterized by the lowest microbial diversity and richness (mean Shannon diversity = 6.64 ± 0.73; mean observed features = 169 ± 80.37).

However, CL2, reflecting the patient-related areas, had the highest microbial diversity, evenness, and richness (mean Shannon diversity = 7.37 ± 1.07, q = 0.0057; mean observed features = 286.81 ± 140.92, q = 0.04; mean Pielou's evenness = 0.93 ± 0.06, q = 0.0057), followed by CL3, the non-patient-related areas (Shannon diversity = 6.72 ± 1.35; observed features = 219.38 ± 133.55; Pielou's evenness = 0.9 ± 0.07) (Fig. 1A).

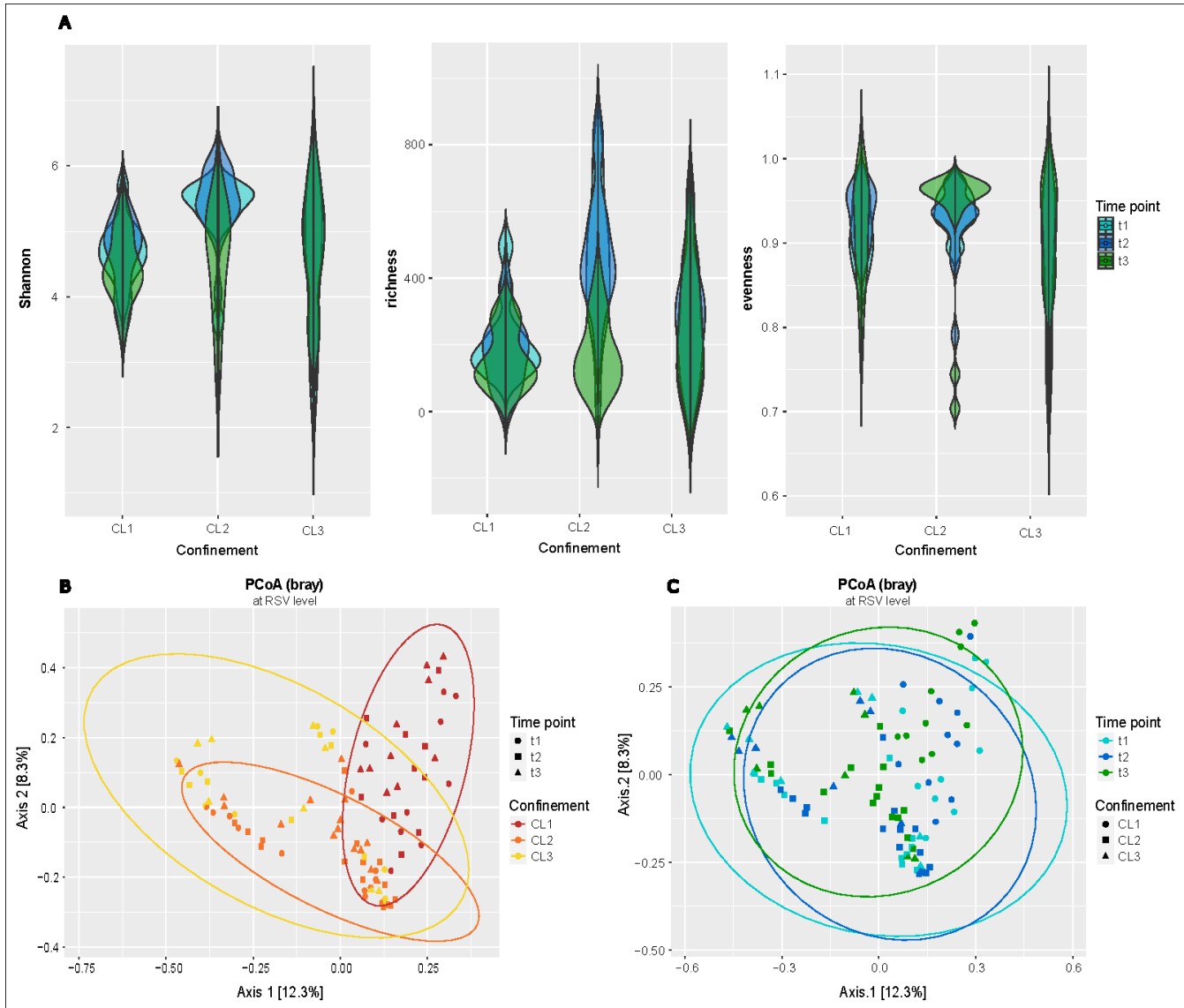

**FIG 1** Confinement level (CL) comparison in regard to alpha and beta diversity of amplicon data on RSV level including longitudinal observations. The statistical output can be found in Table S3. (A) Comparison of alpha diversity indices (Shannon, richness, and evenness) for all CL at all time points (color; t1 - turquoise, t2 - blue, and t3 - green). (B) PCoA based on Bray-Curtis indices of all three CL (color; CL1 - red, CL2 - orange, and CL3 - yellow) and at all three time points (shape; t1 - circle, t2 - square, and t3 - triangle). (C) PCoA based on Bray-Curtis indices for all three time points (color; t1 - turquoise, t2 - blue, and t3 - green). Samples obtained from different CL are represented by shape (CL1 - circle, CL2 - square, and CL3 - triangle).

Likewise, microbial communities differed significantly between all three CLs (PCoA (principal coordinates analysis) based on Bray-Curtis indices) (Fig. 1B). According to MaAsLin2, the highest significance for dissimilarities was observed between samples of CL2 and CL3 (mean Bray-Curtis distance q = 5.6*10E-7), followed by distances of samples between CL1 and CL2 (mean Bray-Curtis distance q = 0.018) and CL1 vs. CL3 (mean Bray-Curtis distance q = 0.04).

Longitudinally, the microbiome diversity and composition remained relatively stable (Fig. 1C). Although we did observe a significant decrease in overall microbial diversity, evenness, and richness over time (MaAsLin2, mean Shannon entropy q = 0.02, mean Pielou's evenness q = 0.02, and mean observed features q = 0.05), the given model coefficient values (effect size) indicate a stronger influence of CL rather than the sampling time point. Additionally, despite observing a significant increase in microbial dissimilarity over time (mean Bray-Curtis distance q = 1.13*10E-8), the coefficient for time

points was comparatively low (coef = 0.006) when compared with those obtained for CLs in terms of beta diversity. A more detailed statistical output for both alpha and beta diversity is available in Table S3.

## Confinement levels are characterized by specific key taxa

We used supervised machine learning models to identify the most predictive features (taxa) for different confinement categories irrespective of their sampling time point. Sample classifications on genus level achieved satisfying performances (see Fig. S3; overall classification accuracy 0.8; CL1 AUC = 1.00, CL2 AUC = 0.99, and CL3 AUC = 0.91) and unveiled *Pseudomonas* (0.11) and *Achromobacter* (0.09) as most predictive features for CL1 and *Stenotrophomonas* (0.09) and *Acinetobacter* (0.06) as most predictive features for CL3 within our trained model.

Differential abundance tests based on ANCOM, ANCOM2 (random intercept model), ANCOM-BC, ALDEx2, and MaAsLin2 were conducted to pinpoint specific key taxa for each CL. Those taxa that showed significant differential abundance in these tests were identified as key taxa representing a particular CL.

Similarly, as already shown for our sample classifications, the differential abundance analysis revealed that *Pseudomonas* (Fig. 2A) and *Achromobacter* (Fig. 2B) were significantly more abundant in CL1, whereas *Acinetobacter* (Fig. 2C) and *Stenotrophomonas* (Fig. 2D) were more abundant in CL3 samples. However, after correcting *P*-values for multi-hypothesis testing, MaAsLin2 did not maintain *Pseudomonas* and *Stenotrophomonas* as significant putative biomarkers for one of the CL, although they were significantly differential abundant with all other tests. For a summary of all differential abundance test results for the identified key taxa, please refer to Table S4. A detailed taxonomic profile is available on our Github repository.

## Highest functional diversity in CL1 is accompanied by the lowest microbial diversity

Performing metagenomic sequencing was challenging due to the required DNA concentration for successful library construction. Therefore, we focused only on floor samples for this analysis and pooled CL1 samples across all time points.

The results showed a similar trend as observed for the amplicon-based analyses, reflected by significantly lower microbial diversity in CL1 compared with CL2 (MaAsLin2, Shannon q = 0.045) and CL3 samples (MaAsLin2, Shannon q = 0.037). No significant differences in microbial alpha diversity were found for CL2 and CL3 (MaAsLin2, Shannon q = 0.47) (Fig. S4A).

Moreover, we determined a significantly higher diversity of overall functions in CL1 compared with CL2 (MaAsLin2, Shannon q = 0.0024) and CL3 samples (MaAsLin2, Shannon q = 0.041). Furthermore, CL3 showed significantly higher functional diversity than CL2 samples (MaAsLin2, Shannon q = 0.0037) (Fig. S4C).

In terms of beta diversity, the metagenomic data also showed significantly different microbial composition of CL1 microbiomes compared with CL2 and CL3 (MaAsLin2, Bray-Curtis distance CL3 q = 0.031; CL2 q = 0.037). However, CL2 and CL3 did not differ significantly (MaAsLin2, Bray-Curtis distance q = 0.46) (Fig. S4B).

These differences were fully reflected by the functional profile, with CL1 samples having a significantly different functional compositions compared with CL2 (MaAsLin2, Bray-Curtis distance q = 0.0019) and CL3 samples (MaAsLin2, Bray-Curtis distance q = 0.002) (Fig. S4D). However, there was also no significant difference in the overall functional profile between samples from CL2 and CL3 (MaAsLin2, Bray-Curtis distance q = 0.25). A summary of the statistical tests is shown in Table S5, and more details can be found on the Github repository.

The microbial profile revealed that the significantly higher abundant key taxa *Acinetobacter* for CL3 and *Pseudomonas* for CL1, which were detected in the 16S rRNA gene amplicon data set, were also among the top 15 most abundant taxa of the

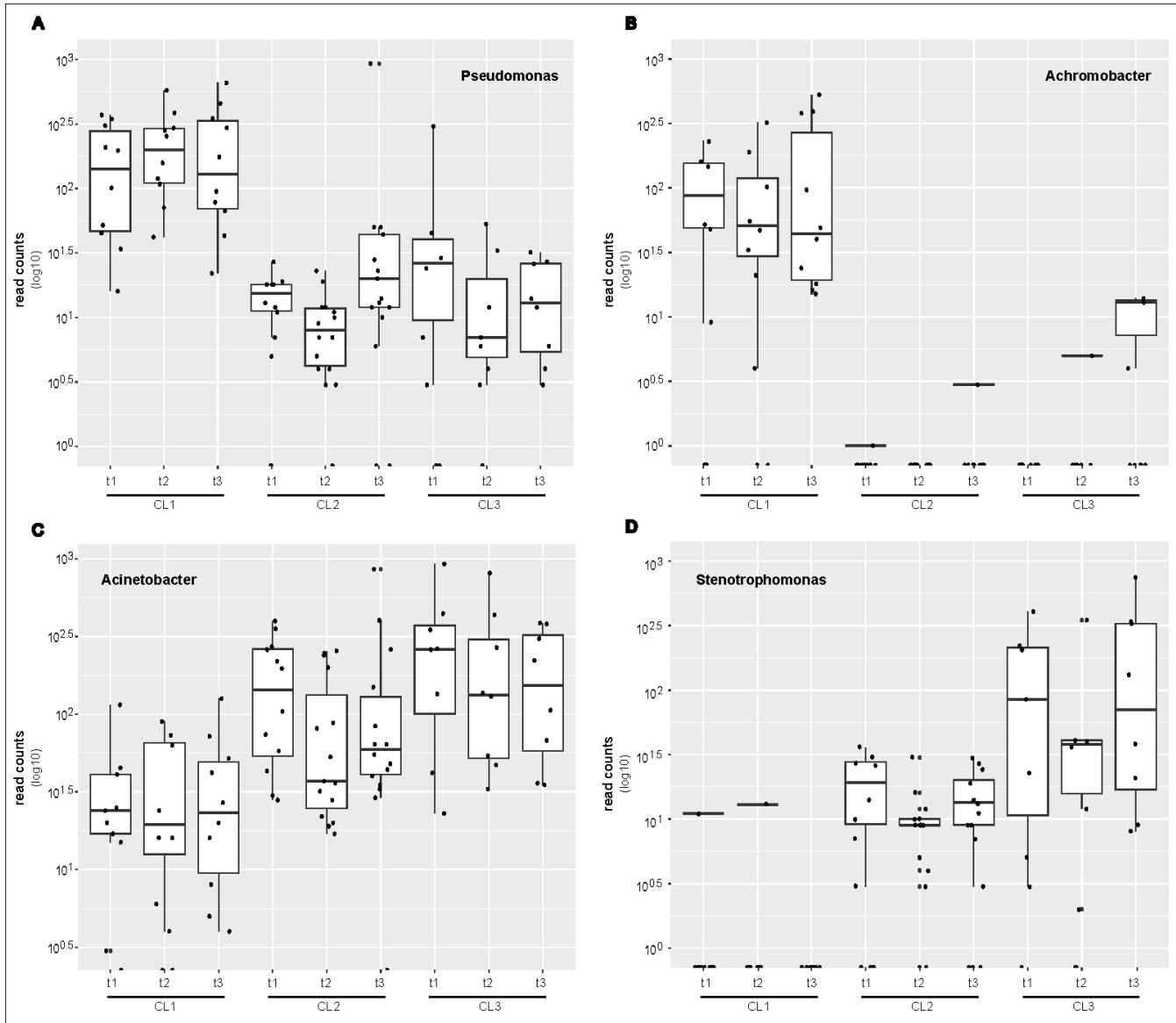

**FIG 2** Relative abundance of key taxa identified in a certain confinement level (CL) obtained from the amplicon data. (A-D) Boxplots for comparison of relative abundance of *Pseudomonas*, *Achromobacter*, *Acinetobacter,* and *Stenotrophomonas* in the three different CLs and at all three time points. Statistics were performed with multiple differential abundance tests, and the results are summarized in Table S4.

metagenomics data set and were also higher abundant in the same CL. Although *Stenotrophomona*s and *Achromobacter* were not among the top 15 most abundant taxa in the metagenomic data, they also showed the same distribution as observed in the amplicon data (Fig. S4E through H; see Github repository), emphasizing their potential role as biomarkers for certain CL.

## Confinement levels differed for key functions related to virulence, disease, defense, and stress response

Due to previous findings which stated that more confined areas showed higher diversities of functions related to virulence (V), virulence, disease, defense (VDD), stress response (SR), general stress response, and stationary phase response (GSR) (5), we analyzed the differences between these functional groups for all CLs. CL1 showed the highest abundance of functions related to all of these groups. CL2 had a higher abundance of functions related to VDD and GSR compared with CL3, whereas functions

related to V and SR were more abundant in CL3 compared with CL2 (Fig. 3A). Nevertheless, according to MaAsLin2, no significant differences could be determined between all CL after *P*-value correction.

To further investigate the functional differences, the top 10 most abundant functions belonging to V, VDD, SR, and GSR were analyzed for each CL. This resulted in 18 key functions that were among the top 10 most abundant functions found either in all CLs, in CL2 and CL3, or in a specific CL. Although some functions were highly abundant in all CLs, others were more specific to a particular CL. CL1 samples especially showed a high abundance of functions related to multidrug efflux pumps.

A more detailed picture of the obtained functions can be found on the Github repository.

## Signatures of *Stenotrophomonas* and *Pseudomonas* are associated with key virulence and stress-response functions

To see if there was a correlation between the 18 selected functions and our observed key taxa, a Spearman's rho correlation was performed for CL2 and CL3 on metagenomic level. Since we only had one pooled sample for CL1, it was not included in the analysis. *Achromobacter* was also not included, since it was mainly present in the excluded sample from CL1.

In CL2, we found a positive correlation of *Acinetobacter* with carbon starvation protein A (cstA) and a negative correlation with the universal stress protein (usp) family. In addition, *Stenotrophomonas* showed a positive correlation to almost all functions within the virulence group. In contrast, *Acinetobacter* in CL3 showed mainly negative correlation to almost all selected functions, with the exception of rubrerythrin and the usp family, for which we found a positive correlation. Interestingly, *Pseudomonas* and *Stenotrophomonas* showed exactly the opposite results with respect to correlations found for *Acinetobacter*. However, after correction for the *P*-value, we found no significance for the Spearman's rho correlation of the key taxa and selected functions (Fig. 3B).

## Key resistance, virulence, and stress functions are encoded in assembled genomes from all CLs

From the shotgun metagenomics data, we reconstructed contigs to 25 metagenome-assembled genomes (MAGs) with >90% completeness and <25% contamination. Most MAGs were obtained from CL3 (13 MAGs), followed by CL2 (11 MAGs) and CL1 (1 MAG). The obtained MAGs are available on the Github repository. To compare the MAGs from the different CLs, a protein catalog was created based on the eggNOG mapper results (exported from MaGe) and weighted by the mean relative abundance (calculated with coverM) of each MAG per CL. The protein catalog from the genome-centric functional annotation showed the same trend in alpha diversity as the gene-centric data, with CL1 showing the highest alpha diversity, followed by CL3 and CL2. However, beta diversity of the protein catalog showed no differences for the MAGs of the different CLs (see Github repository).

Additionally, we wanted to see whether the selected functions from the gene-centric data belonging to V, VDD, SR, and GSR were also present in our obtained MAGs. Therefore, we examined the presence of those functions that could be assigned to a COG ID using the EggNOG Database in relation to the obtained MAG and CL. The selected functions (of which an assignment was possible) were present in all CL and all MAGs, except for COG 3965 (predicted Co/Zn/Cd cation transporter, for the selected function cobalt-zinc-cadmium resistance protein CzcD), which was not present in *Flavobacterium* of CL2 and one of two MAGs from *Cutibacterium* of CL3 (see Github repository). Moreover, no dissimilarities were detected among MAGs obtained from the different CLs.

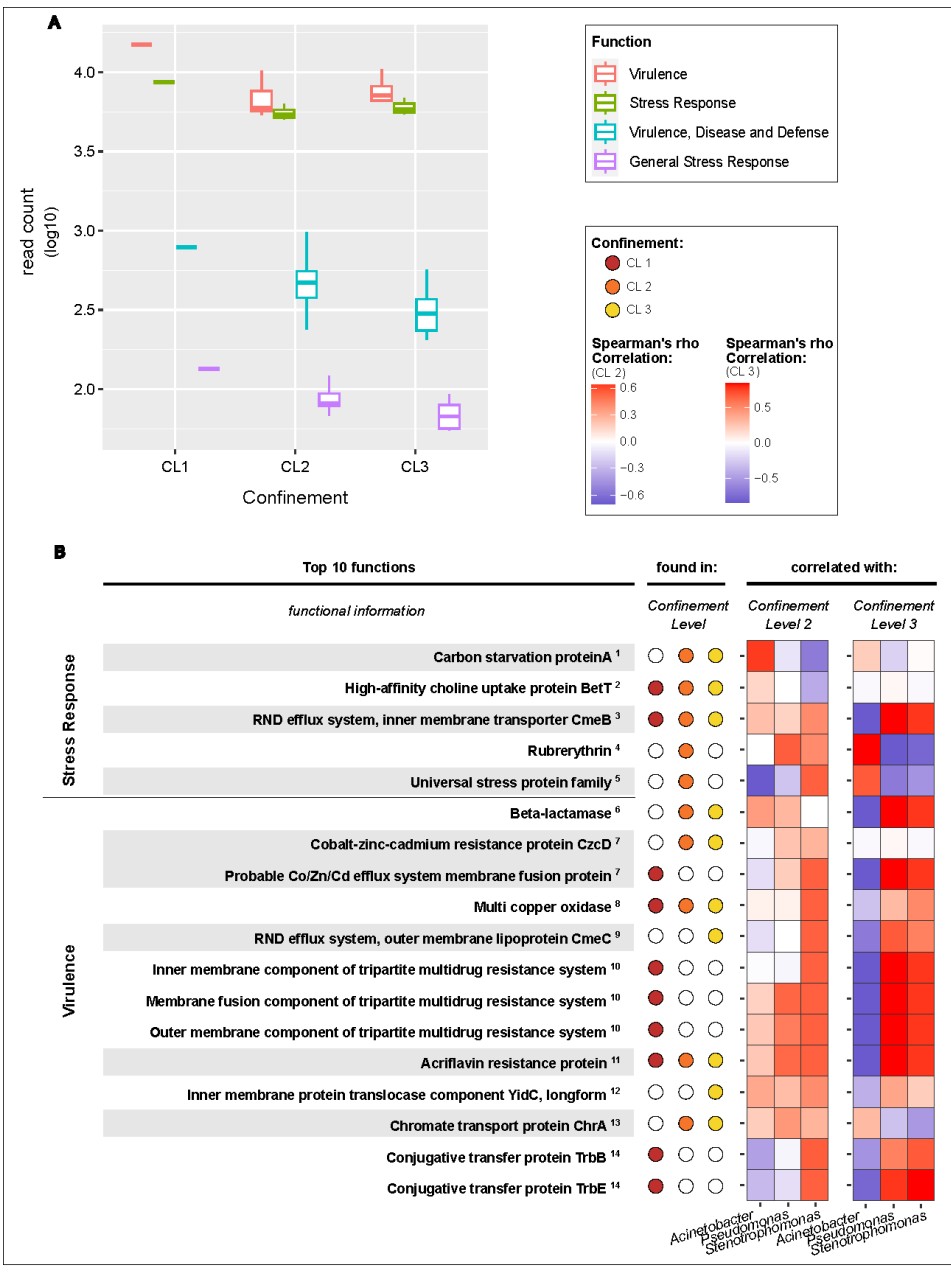

FIG 3 Functional comparison of the different confinement levels (CLs). (A) Relative abundance of metagenomic functions related to virulence (red), virulence, disease, defense (blue), stress response (green), and general stress response (magenta) for

**FIG 3** (Continued)

all CLs. (B) Spearman's rho correlation of the observed key taxa with the most abundant functions related to virulence and stress response. Red color displays a positive correlation, and blue, a negative correlation. Filled dots represent the presence of the function in a certain CL (CL1 - red, CL2 - orange; CL3 - yellow).

## Confinement levels are related to certain antibiotic resistance types

To obtain information on resistance genes, we profiled high-quality MAGs according to the Comprehensive Antibiotic Resistance Database (CARD). The MAG of *Pseudomonas* obtained from CL1 showed almost all antibiotic resistances acquired via resistance-nodulation-cell division (RND) antibiotic efflux pumps, with the exception of fluoroquinolone, macrolide, phenicol antibiotic, and tetracyclines. MAGs obtained from the other CLs were not represented in the group of RND class of antibiotic resistances or were represented only at specific time points. However, MAGs of CL2 and CL3 showed resistance to carbapenem, cephalosporin, and penam obtained via OXA beta-lactamase from antibiotics inactivation-derived resistance at all time points. Resistance from this group was not found in the MAG of CL1, also not from antibiotic target alteration, where CL3 assigned MAGs showed resistances to lincosamide, macrolide, and streptogramin obtained via Erm 23S ribosomal RNA methyltransferase at all three time points (Fig. 4A and with sampling time point information, see Github repository).

## Resistance genes encoded on plasmids or chromosomes follow CL classifications

Resistances carried on plasmids are of great interest as they can be transferred by horizontal gene transfer. Therefore, we profiled additionally our contigs according to the CARD and differentiated the results into resistances potentially resulting from chromosomes or plasmids with PlasFlow.

CL1 metagenomes showed resistance to fluoroquinolone and macrolide antibiotics carried on plasmids and to classes of lincosamides, macrolides, and streptogramins on both plasmids and chromosomes. For this class of antibiotics, resistances were also observed for CL2 present on plasmids and for CL3 also both on plasmids and chromosomes at all time points. In general, resistances to fluoroquinolones present on plasmids were also found in all CLs and at all time points (Fig. 4B).

The resistance profiles of CL1 samples predominantly comprised antibiotic resistances obtained through antibiotic efflux, accounting for 75% of all resistances detected in CL1 samples. Moreover, these resistances were primarily located on chromosomes. However, resistances belonging to this category were also highly abundant in samples from other CLs (44% for CL2 and 40% for CL3), albeit not at all time points. Antibiotic efflux-mediated resistances were found to be predominantly located on chromosomes across all CLs (89% for CL1, 73% for CL2, and 73% for CL3).

Regarding plasmid-mediated antibiotic resistance, CL1 exhibited the lowest proportion of resistances (25%), followed by CL3 (48%), and the highest proportion of resistances was observed in CL2 (56%). The reported percentages are based on the total resistances detected for each CL. Considering the concerning spread of antibiotic resistance via plasmids, these findings highlight the variation in the extent of plasmid-mediated antibiotic resistance among the different CLs. Additionally, antibiotic inactivation resistance was primarily detected in CL2 (accounting for 38% of all resistances observed in this CL), followed by CL3 (33%) and CL1 (8%).

However, resistances pertaining to this category were found on plasmids for all CLs (100% for CL1, 69% for CL2, and 67% for CL3) but only at specific time points. Notably, CL1 exhibited only one resistance in this group, which was located on plasmids.

Samples from CL2 contained resistances to lincosamide, macrolide, oxazolidinone, phenicol antibiotic, pleuromutilin, streptogramin, and tetracycline acquired through antibiotic target protection, which was detected on plasmids at all time points and in CL3 at one time point. Generally, resistances obtained via antibiotic target protection and

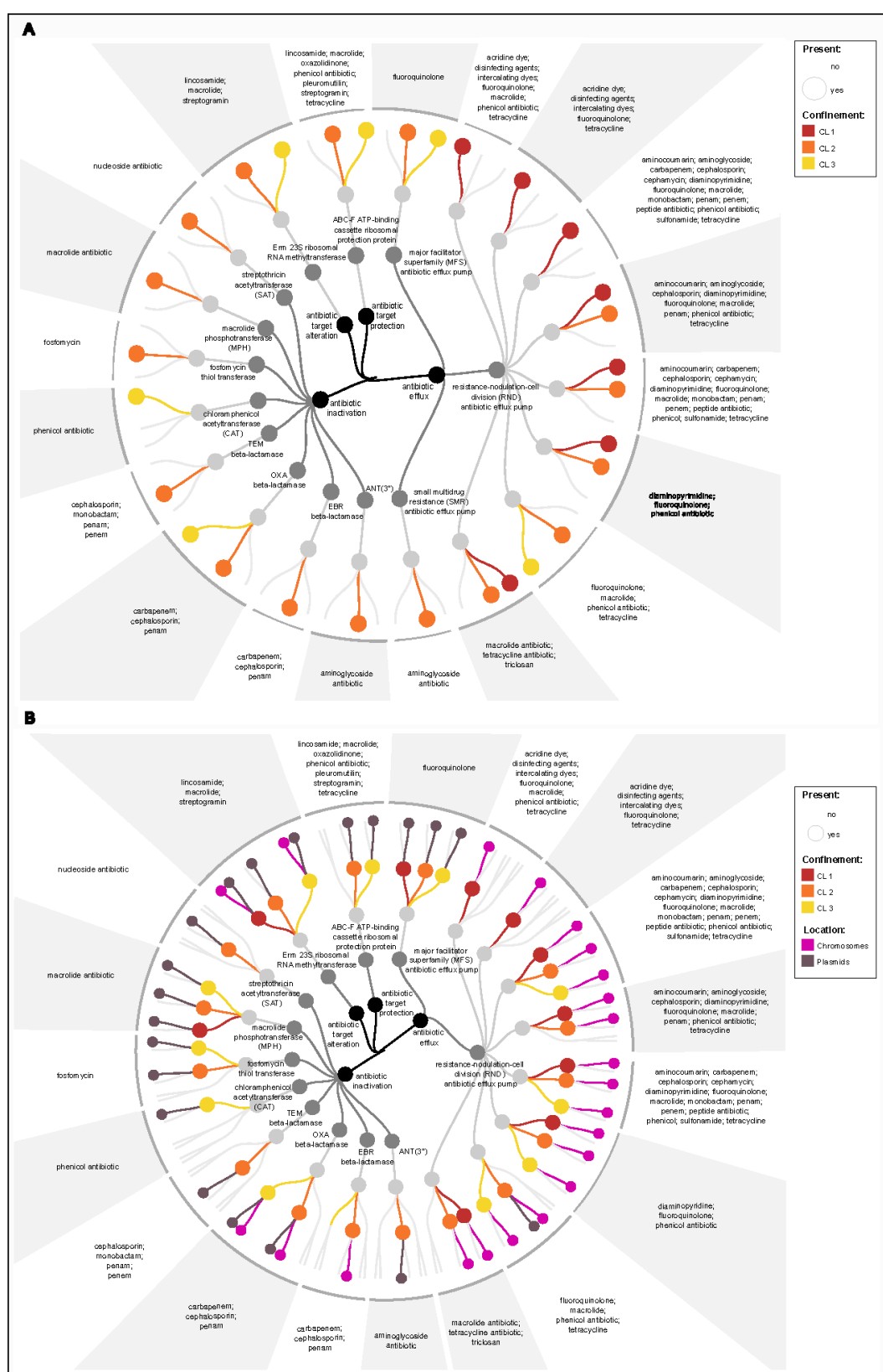

**FIG 4** Comparison of antibiotic resistances found in the different confinement levels (CLs). (A) Cladogram of antibiotic resistances profiled with high-quality MAGs according to the CARD. (B) Cladogram of antibiotic resistances profiled with contigs according to CARD and differentiated resistances potentially resulting from chromosomes or plasmids with PlasFlow. Dots are showing the presence of certain antibiotic resistance in a given CL (CL1 - red, CL2 - orange, and CL3 - yellow). Pink dots show resistances present on chromosomes and violet dots on plasmids.

alteration were more frequently detected on plasmids rather than chromosomes across all CL (with 50% in CL1, 100% in CL2, and 75% in CL3 of obtained resistances found on plasmids in the specific CL). For further details and specific information on individual time points, please refer to our Github repository.

## GRiD scores and PMA-treated samples indicate viability of resistance-bearing bacteria in all CLs

Actively growing microorganisms are of specific concern in confined areas of a hospital environment. We used GRiD (Growth Rate InDex) to estimate the replication rates of our MAGs to investigate differences in growth rates within the different CLs. In addition, the results were compared with the amplicon data of the PMA-treated samples, which should also display intact, potentially active microbes. No differences in growth rates were observed between CLs. However, only GRiD values >= 1 were observed in CL1.

*Pseudomonas*, the only obtained MAG in CL1, was also found to be highly abundant in the PMA amplicon data, also suggesting that this genus was potentially alive at the time of sampling (Fig. S5, Github repository).

In CL2 and CL3, some MAGs had GRiD values of zero, and based on (41) GRiD scores below 1.02 are considered non-replicating. However, a high variance of GRiD values above one was observed in CL3. Overall, MAGs classified as *Pseudomonas* (mean GRiD value = 1.28), *Acinetobacter* (mean GRiD value = 1.15), and *Cutibacterium acnes* (mean GRiD value = 1.11) showed the highest GRiD values, whereas the other MAGs had only GRiD values below 1.02. Furthermore, GRiD values above 1 and with a high variance were observed for *Acinetobacter*, with differences in replication between CL2 and CL3 for this genus (*Acinetobacter* CL2, mean GRiD value = 1; *Acinetobacter* CL3, mean GRiD value = 1.22). This coincides with PMA-treated samples, where *Acinetobacter* showed the highest abundance in CL3. *Cutibacterium acnes* showed a higher growth rate in CL2 (mean GRiD value = 1.35) in comparison to CL3 (mean GRiD value = 0.75), whereas Xanthomonadaceae showed higher growth rates in CL3 (mean GRiD value = 1.03) than in CL2 (mean GRiD value = 0.227). In the PMA-treated amplicon samples *Cutibacterium* was also found to be higher abundant in CL2 and Xanthomonadaceae in CL3 (Fig. S5), indicating that these genera were potentially alive in these CL at the time of sampling. All GRiD values and PMA data are available on Github repository.

## DISCUSSION

Understanding the hospital microbiome is crucial for developing effective strategies to prevent and control healthcare-associated infections (HAIs) and the spread of antimicrobial resistance (AMR). It is particularly important to investigate factors that have an impact on hospital microbiome and resistome to improve our understanding of microbial dynamics and therefore ensure patient's recovery.

We analyzed the microbiome and resistome in different aspects (summarized in Table 1) and with a longitudinal study design, namely on one hand with amplicon data (covering various surfaces and rooms), and on the other hand with shotgun metagenomic data using two approaches: (i) read-based metagenome and (ii) a genome-based metagenome analysis. Besides genome analysis, we performed plasmid reconstruction, investigated antimicrobial resistances, and estimated growth rates.

Amplicon and shotgun metagenomic sequencing revealed significant differences in microbial and functional compositions between confinement levels (CLs) within a single hospital. The most confined area, CL1 (the operating room), showed the lowest microbial diversity but the highest functional diversity compared with other CLs. These results support previous findings that increased confinement leads to decreased microbial diversity and increased resistance gene diversity (5).

Other studies have also observed a decline in bacterial diversity, a shift in microbial composition, and increased resistances due to confinement, indicating its impact on the microbial profile, and function, and that confinement promotes the spread of antibiotic resistance genes in environmental microbiomes (5, 6, 56).

**TABLE 1** Summary of main results. It includes information about diversity changes, observed key taxa, obtained MAGs, found antibiotic (AB) resistances, if they are carried more likely on plasmids or chromosomes for each confinement level (CL), and information of potentially alive microbes[a]

| | | CL1 | CL2 | CL3 |
|---|---|---|---|---|
| Amplicon | Microbial diversity | + | +++ | ++ |
| | Obtained key taxa | *Pseudomonas Achromobacter* | | *Acinetobacter Stenotrophomonas* |
| Metagenomics | Microbial diversity | + | ++ | +++ |
| | Functional diversity | +++ | + | ++ |
| | V related functions | +++ | + | ++ |
| | VDD related functions | +++ | ++ | + |
| | SR related functions | +++ | + | ++ |
| | GSR related functions | +++ | ++ | + |
| | MAGs obtained | 1 | 11 | 13 |
| | AB resistances obtained | AB efflux: 75%<br>AB target alteration & protection: 17%<br>AB inactivation: 8% | AB efflux: 44%<br>AB target alteration & protection: 18%<br>AB inactivation: 38% | AB efflux: 41%<br>AB target alteration & protection: 26%<br>AB inactivation: 33% |
| | AB resistances (Chromosome or Plasmid) | Chromosome: 75%<br>Plasmids: 25% | Chromosome: 44%<br>Plasmids: 56% | Chromosome: 52%<br>Plasmids: 48% |
| | GRiD (replicating microbes) | *Pseudomonas* | *Cutibacterium acnes*<br>Bacteria | *Acinetobacter Xanthomonadaceae* |

[a]Whenever a decrease in diversity was observed, the corresponding cell is marked with a plus sign. On the other hand, when an increase in diversity was found, the cell was marked with three plus signs. If the change in diversity falls somewhere in between, the cell is marked with two plus signs. Percentages given are based on the total resistances obtained for each CL.

To investigate the stability of the hospital microbiome, we performed longitudinal observations and found that the microbial profile remained relatively stable over time covering the same microbial taxa as the most important predictors for CL1 and CL3 within our supervised machine-learning models. This stability could be attributed to the limited microbial exchange and environmental impact in these confined settings (6).

Moreover, certain key taxa were found to be specifically present in a particular CL, with a significantly higher abundance of more environmentally associated genera *Acinetobacter* and *Stenotrophomonas* in samples of the least confined areas (CL3) and *Pseudomonas* and *Achromobacter* in samples of the most confined area (CL1), the operating room.

*Achromobacter* is commonly isolated from the respiratory tract and has been detected also in contaminated solutions, water, and even disinfectants (57–60). As samples from CL1 were obtained in an operating room used for thoracic surgery, patients could be considered a potential source. *Pseudomonas* can survive on different surfaces and was reported to show a major predominance in hospital microbiomes (20). It was also described in the water supply of hospitals (61–63), leading to possible contamination of various surfaces in the operating room. In sink samples from CL2, *Pseudomonas* was highly abundant, consistent with previous reports (61–63).

However, sink samples from CL3 had a higher abundance of *Stenotrophomonas* and *Sphingobium*. *Stenotrophomonas* is commonly found in water samples (64), and *Acinetobacter* and *Stenotrophomonas* are frequently detected in hospital environments and on different surfaces, including medical equipment (20, 64–66). *Acinetobacter* can persist on various surfaces for several months, making environmental contamination difficult to control (67). Although no significantly higher abundant taxa were detected in samples from CL2, a higher abundance of human-skin-associated microorganisms, such as *Corynebacterium* and *Staphylococcus*, was observed in this CL.

As confinement was previously described with an increase in resistance gene diversity and an increase in opportunistic pathogens (5), we further investigated functions related to V, VDD, SR, and GSR. We found the overall highest abundance of functions related to these groups in CL1, followed by CL2 for VDD and GSR and CL3 for functions related to V and SR.

V- and VDD-related functions are associated with higher exposure to certain stressors such as antibiotics, disinfectants, and other biocidal agents or heavy metals (5). However, functions related to these groups are also increased in bacteria that have the ability to form biofilms, like the observed key taxa (*Pseudomonas*, *Achromobacter*, *Acinetobacter*, and *Stenotrophomonas*) (60, 68–70). Besides that, bacteria with functions associated with VDD may upregulate genes involved in adhesion, invasion, and toxin production. This upregulation allows the bacteria to better colonize and cause disease in the host (71–73). CL1 displayed a higher abundance of functions related to opportunistic pathogenic microbes, suggesting that it may serve as a potential source for these microorganisms that can survive in a highly maintained and restricted environment. In CL2, functions associated with VDD were more abundant, possibly due to the presence of patients. On the other hand, V- and SR-associated functions were more abundant in CL3, which may be attributed to the presence of more selective stressors in this environment.

To get a more detailed picture, we further examined the top 10 most abundant functions found in all CL, which were associated with V and SR, and correlated those with our observed key taxa.

In CL3, *Acinetobacter* showed a negative correlation with V-associated functions, except for chromate transport protein *Chr*A, and a positive correlation with SR-associated functions such as rubrerythrin and universal stress protein family. Rubrerythrin helps defend against oxidative stress, for example, caused by hydrogen peroxide, a common ingredient in cleaning agents (74). The universal stress protein family also plays a crucial role in protecting *Acinetobacter* from various stressors including hydrogen peroxide and low pH (75).

*Acinetobacter* is known as an opportunistic pathogen associated with healthcare-associated infections, and universal protein A was previously suggested as a potential target for new therapeutics against *Acinetobacter* (75).

In contrast to *Acinetobacter*, *Stenotrophomonas* and *Pseudomonas* showed mainly positive correlations with V-associated functions and negative correlations with SR-related functions. In these two genera, functions related to multidrug resistance and conjugation were positively correlated. These functions are of great concern, as they lead to antimicrobial resistance spread (76). However, these correlations were weaker in patient-related areas of CL2 compared with CL3. Results for CL1 were also not available due to missing replicates, and *Achromobacter* was not included due to its predominance in the excluded CL1.

Due to the fact of highly abundant functions leading to multidrug resistances, we examined the antibiotic resistances in the different CLs and investigated their presence on chromosomes or plasmids. This is of great interest as resistances found in plasmids can be transferred via horizontal gene transfer and therefore promote the spread of antibiotic resistances (77).

In the ICU commonly used antibiotics are piperacillin/tazobactam, ampicillin/sulbactam, meropenem, linezolid, daptomycin, moxifloxacin, levofloxacin, and cefepime. These antibiotics belong to the antibiotic classes of acyl-aminopenicillins (piperacillin/tazobactam), beta-lactamase antibiotics (ampicillin/sulbactam), carbapenem (meropenem), oxazolidinone (linezolid), cyclic lipopeptides (daptomycin), fluoroquinolone (moxifloxacin, levofloxacin) and cephalosporins (cefepime) (42).

Resistances against fluoroquinolone were observed in all three CLs, mainly on chromosomes, except for those associated with antibiotic target protection, which were found on plasmids across all CLs. Carbapenem and cephalosporin resistances were found on both chromosomes and plasmids in all CL, whereas oxazolidinone resistances were only observed on plasmids in CL2 and CL3. No resistances were found against other antibiotic classes used in the ICU. However, also resistances to other antibiotics were observed. Generally, a higher prevalence of resistances on chromosomes was found in CL1 and CL3, whereas CL2 had more resistances on plasmids. This could be particularly important because in CL (patient-related areas), the potential spread of antibiotic resistance could be a major concern.

Especially actively growing, viable microbes are of great concern in hospital environments, also with respect to their ability to infect patients and endanger their recovery. We found actively growing bacteria in all CL, with the highest estimated growth rates for CL1 (operating room), followed by CL3 (non-patient-related areas) and CL2 (patient-related areas). We also observed growth rates for the observed key taxa (*Pseudomonas*, *Acinetobacter*, *Xanthomonadacae/Stenotrophomonas*), except for *Achromobacter*. The obtained results could also be confirmed by the PMA-treated amplicon data.

Although our study analyzed the microbiome and resistome across different CLs in a single hospital, there are limitations to our analysis. We only focused on floor samples for shotgun metagenomics due to the low biomass and pooled samples across all time points of CL1 for successful library preparation, which limits the representativeness of subsequent analysis and statistical tests.

Additionally, we lacked information on cleaning and monitoring procedures for each CL, which could have influenced the hospital microbiome and resistome. To obtain a better understanding of the dynamics of the microbiome and resistome within hospital settings, future studies should sample more similar rooms within the same CL and across different hospitals. Including a diverse range of hospitals, especially those with varied cleaning practices and from different countries, would help mitigate confounding variables specific to a single hospital environment. Moving forward, a comparative analysis across various hospitals, potentially using a meta-analysis, could more definitively ascertain the influence of CL on the hospital microbiome and resistome. Therefore, further studies are needed to validate our findings.

One strategy to mitigate the loss of microbial diversity in hospitals is to manipulate the microbiome. *Bacillus* spores have shown potential in reducing harmful microbes and antimicrobial resistance in hospital settings (1, 78). Introducing beneficial microbes could reduce the prevalence of infection-causing microbes and potentially change the resistome, providing better protection for patients. However, in operating rooms, other methods could be explored, such as targeting functions important for the survival of opportunistic pathogens (75). Additionally, the overall use of antimicrobial substances should be carefully considered, as we have no sufficient alternative treatment. As the hospital microbiome is directly influenced by patients and vice versa, it is important to better understand the dynamics of microbiomes and resistomes within a hospital setting. Therefore, more studies are needed in the future to develop new strategies for restoring microbial diversity and reducing the spread of antimicrobial resistances to ensure a safe patient recovery.

## ACKNOWLEDGMENTS

The authors would like to acknowledge the computational resources of the MedBioNode at the Medical University of Graz and the support of the ZMF team at the Core Facility Computational Bioanalytics (Medical University of Graz), especially Marija Durdevic, MSc. for her advice on statistics. The authors would also like to thank the Core Facility Molecular Biology (ZMF, Medical University of Graz) for library preparation and sequencing. The authors also appreciate the input and support of Univ.-Prof. Dr. med. univ. Robert Krause and Dr. med. univ. Philipp Eller. The authors also want to thank the Medical University of Graz (Start Förderung) for funding this project.

The study was designed by K.K. Sampling was performed by S.D., K.K., and L.W., and samples were processed by S.D. S.D., K.K., and A.M. performed the data analysis, and statistical tests were performed by S.D. and A.M. S.D. and C.K. created all shown figures. S.D. wrote the manuscript, and K.K., C.M.E., C.K., and A.M. contributed to the writing of the manuscript, which was read and approved by all authors.

## AUTHOR AFFILIATIONS

[1]D&R Institute of Hygiene, Microbiology and Environmental Medicine, Medical University of Graz, Graz, Austria
[2]BioTechMed Graz, Graz, Austria

## AUTHOR ORCIDs

Stefanie Duller [ID] http://orcid.org/0000-0002-0927-3270
Christina Kumpitsch [ID] http://orcid.org/0000-0002-2077-2839
Alexander Mahnert [ID] http://orcid.org/0000-0001-7083-8894

## FUNDING

| Funder | Grant(s) | Author(s) |
|---|---|---|
| Medizinische Universität Wien (MediUni Wien) | | Christine Moissl-Eichinger |

## DATA AVAILABILITY

The raw amplicon and shotgun metagenomics data can be found at: European Nucleotide Archive (ENA) under project ID: PRJEB60571 and the processed data, the metadata (with ENA Accession number), the feature table, all differential abundance tests, the output from abricate, the taxonomical and functional outputs from MEGAN, the GRiD values, all good quality MAGS the STORMS (Strengthening The Organizing and Reporting of Microbiome Studies) checklist (79) as well as respective scripts and commands to reproduce our analyses in detail are available on our Github repository: https://github.com/SDMUG/Hospital_Resistome.

## ADDITIONAL FILES

The following material is available online.

Supplemental Material

**Supplemental material (mSystems00726-24-s0001.pdf).** Supplemental tables and figures.

Open Peer Review

**PEER REVIEW HISTORY (review-history.pdf).** An accounting of the reviewer comments and feedback.

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
