## [Reviewer comments · mSystems]

In-hospital areas with distinct maintenance and staff- /patient traffic have specific microbiome profiles, functions, and resistomes

Stefanie Duller, Christina Kumpitsch, Christine Moissl-Eichinger, Lisa Wink, Kaisa Koskinen, and Alexander Mahnert

Corresponding Author(s): Alexander Mahnert, Medizinische Universitat Graz

Review Timeline:

Submission Date:

May 29, 2024

Accepted:

June 11, 2024

Editor: John Gibbons

Reviewer(s): Disclosure of reviewer identity is with reference to reviewer comments included in decision letter(s). The following individuals involved in review of your submission have agreed to reveal their identity: Scott T Kelley (Reviewer #3)

Transaction Report:

DOI: <https://doi.org/10.1128/msystems.00726-24>

Re: mSystems00726-24 (In-hospital areas with distinct maintenance and staff-/patient traffic have specific microbiome profiles, functions, and resistomes)

Dear Dr. Alexander Mahnert:

Congratulations, and thank you for submitting this interesting manuscript to mSystems!

Your manuscript has been accepted, and I am forwarding it to the ASM production staff for publication. Your paper will first be checked to make sure all elements meet the technical requirements. ASM staff will contact you if anything needs to be revised before copyediting and production can begin. Otherwise, you will be notified when your proofs are ready to be viewed.

Sincerely,
John Gibbons
Editor
mSystems

Reviewer #1 (Comments for the Author):

I appreciate the authors' thoughtful response to my previous notes and believe they have addressed my concerns.

Reviewer #3 (Comments for the Author):

The authors have addressed all my concerns from the previous review. It is a very strong paper that should be valuable to the field.